# Process and Material Analysis of Laser- and Convection-Dried Silicon–Graphite Anodes for Lithium-Ion Batteries

Sebastian Wolf [1,*], Laura Garbade [1], Vinzenz Göken [2], Rebekka Tien [2], Markus Börner [2], Daniel Neb [1] and Heiner Hans Heimes [1]

[1] Production Engineering of E-Mobility Components (PEM), RWTH Aachen University, Bohr 12, 52072 Aachen, Germany; l.garbade@pem.rwth-aachen.de (L.G.); d.neb@pem.rwth-aachen.de (D.N.); h.heimes@pem.rwth-aachen.de (H.H.H.)

[2] Münster Electrochemical Energy Technology (MEET), Westfälische Wilhelms-Universität Münster, Corrensstraße 46, 48149 Muenster, Germany; vinzenz.goeken@uni-muenster.de (V.G.); rebekka.tien@uni-muenster.de (R.T.); markus.boerner@uni-muenster.de (M.B.)

[*] Correspondence: s.wolf@pem.rwth-aachen.de

**Abstract:** Drying electrodes is very cost-intensive as it is characterized by high energy and space consumption. Laser drying is considered a promising alternative process due to direct energy input and lower operating costs. However, it is unclear whether the same product and process quality can be achieved with laser drying. Silicon–graphite anodes with different silicon contents were processed using either a high-power diode laser or a convection oven. The laser-drying process was investigated using thermography, and the effect of laser drying on the electrode quality was examined using adhesion and residual moisture measurements. Furthermore, thermogravimetric analysis, SEM images and electrical conductivity were used to analyse the laser- and convection-dried anodes. It was shown that silicon–graphite anodes can also be manufactured using laser drying, with a significant reduction in drying time of over 80%.

**Keywords:** laser drying; silicon–graphite anodes; electrode manufacturing; lithium-ion battery production



## 1. Introduction

Currently, convection ovens are commonly used to dry electrodes for lithium-ion batteries. Convection drying is the most energy-intensive process step, accounting for 27 to 47% of total energy consumption in battery production [1,2]. As a result of the high energy demand and dryer lengths of up to 100 m at web speeds of 80 m/min, the drying step is responsible for 21% of the CapEx and OpEx of battery production [3,4]. In addition, challenges exist to realize homogeneous drying on the entire electrode width, as the drying process is very sensitive to inhomogeneous temperatures and air flows as well as changes in electrode dimensions such as the wet film thickness. Thus, there is a high interest in supplementing this process step with more energy-efficient alternatives or substituting it completely. Laser drying represents a promising process alternative and significant progress has been made in the implementation of this new drying technology in recent years [5,6]. However, the literature lacks a detailed description of the laser-based drying process and its influence on electrode quality [5].

In addition to the approach of modifying carbon active materials, a combination of silicon and graphite as anode material shows the potential to improve battery cell energy density [7]. For this reason, a trend towards the use of silicon–graphite composites for application in anodes can be identified. The processability of silicon–graphite anodes with laser drying has not yet been investigated. Therefore, laser- and convection-dried silicon–graphite anodes are manufactured to compare the influence on electrode quality.

## 2. Experimental Set-Up

Silicon–graphite anodes with varying silicon content were manufactured in a roll-to-roll coating process and dried by convection or laser. The electrode quality was subsequently examined by looking into thermography, adhesion force, residual moisture, scanning electron microscope (SEM), electric conductivity and thermogravimetric analysis.

### 2.1. Materials

Anode slurries with different silicon–graphite ratios were prepared under protective atmosphere. As active materials, commercial graphite ($d_{particle}$ = 40 µm) and silicon (Silgrain e-Si 410, Elkem ASA, Oslo, Norway, $d_{particle}$ = 3.20 µm) were used. Further slurry components were styrene butadiene rubber (SBR, BM-451B, Zeon Europe GmbH, Dusseldorf, Germany), carboxymethyl cellulose (CMC, MAC500LC, Nippon Paper Industries Co., Tokio, Japan) and carbon black (C-Nergy Super C45, Imerys S.A., Paris, France) as well as deionized water as a solvent. None of the materials underwent any additional purification or pre-blending process before mixing. The different material compositions are listed in Table 1. Mixtures with a 5, 10 and 20% silicon content, in regards of the solid components, were examined. A distinction was made between slurries used for convection and laser drying. In accordance with prior experiments, mixtures for laser drying were composed with a larger solvent content to allow for a higher laser intensity during the drying process without binder destruction.

**Table 1.** Slurry compositions for convection and laser drying of silicon–graphite anodes.

| Composition | | Graphite | Silicon | SBR | CMC | Carbon Black | Solvent |
|---|---|---|---|---|---|---|---|
| 5% Si-Conv. | $w_{solid}$ (%) [1] | 89.00 | 5.00 | 3.00 | 2.00 | 1.00 | 0.00 |
| | $w_t$ (%) [2] | 40.05 | 2.25 | 3.40 | 0.90 | 0.45 | 52.95 |
| 5% Si-Laser | $w_{solid}$ (%) | 89.00 | 5.00 | 3.00 | 2.00 | 1.00 | 0.00 |
| | $w_t$ (%) | 39.29 | 2.21 | 3.31 | 0.88 | 0.44 | 53.91 |
| 10% Si-Conv. | $w_{solid}$ (%) | 84.00 | 10.00 | 3.00 | 2.00 | 1.00 | 0.00 |
| | $w_t$ (%) | 37.80 | 4.50 | 3.40 | 0.90 | 0.45 | 52.95 |
| 10% Si-Laser | $w_{solid}$ (%) | 84.00 | 10.00 | 3.00 | 2.00 | 1.00 | 0.00 |
| | $w_t$ (%) | 37.09 | 4.42 | 3.31 | 0.88 | 0.44 | 53.91 |
| 20% Si-Conv. | $w_{solid}$ (%) | 74.00 | 20.00 | 3.00 | 2.00 | 1.00 | 0.00 |
| | $w_t$ (%) | 33.30 | 9.00 | 3.40 | 0.90 | 0.45 | 52.95 |
| 20% Si-Laser | $w_{solid}$ (%) | 74.00 | 20.00 | 3.00 | 2.00 | 1.00 | 0.00 |
| | $w_t$ (%) | 32.67 | 8.83 | 3.31 | 0.88 | 0.44 | 53.91 |

[1] Mass percentage of solid components. [2] Total mass percentage, including liquid and solid components.

The mixing process was conducted in an intensive mixer with a rotating mixing container with double speed and a mixing tool (EL 1, Maschinenfabrik Gustav Eirich GmbH & Co. KG, Hardheim, Germany). First, graphite, silicon, carbon black and CMC were mixed at 350 rpm for 15 min and subsequently at 500 rpm for 5 min to ensure uniform mixture of the solids. In another container, the solvent and SBR were mixed at 300 rpm for 5 min to obtain a homogeneous solution of the liquid components. A quarter of the dry mixture was then added to the solution and mixed at 1050 rpm for 5 min. This process was repeated three times until all dry components were mixed into the liquids. To achieve a high homogeneity, the slurry was then mixed at 2000 rpm for 15 min and additional 45 min. Between each mixing step, the side walls of the container and the mixing tool were scraped off to avoid the formation of agglomerates. In a final mixing step, the slurry was blended at 300 rpm for 60 min to reduce air bubbles within the solution. The temperature of the final slurry was approximately 40 °C. The mixing procedure was kept the same for all material compositions.

## 2.2. Set-Up and Process Parameters

Immediately after mixing, the slurries were coated and dried on one side of a copper foil (Avocet Steel Strip Ltd., $t_{foil}$ = 10 μm) in a roll-to-roll process. The experimental set-up of the coating and drying process is visualized in Figure 1.

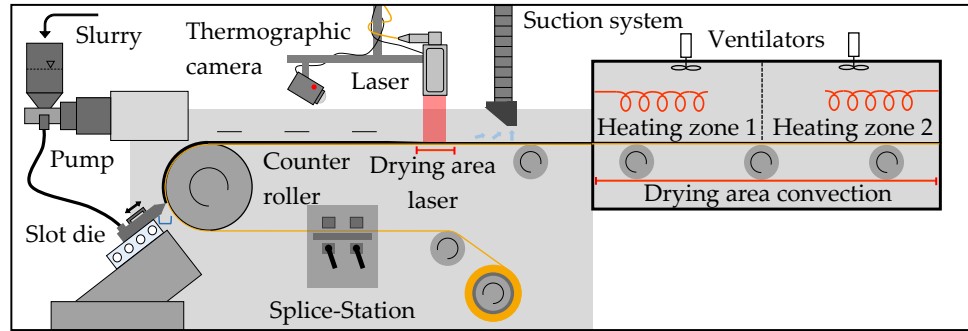

**Figure 1.** Schematic set-up of the coating and drying process with laser and convection [6].

A slot die coating system (Robert Bürkle GmbH, Freudenstadt, Germany) with adjustable web speed was used. The material flow rate of the slurry was regulated by the speed of the pump. The distance from slot die to copper foil was 200 μm. In the subsequent drying process, either a laser system or a convection dryer was used. The laser system consisted of a diode laser (LDM 8000 W, Laserline GmbH, Mülheim-Kärlich, Germany) with a wavelength range of 900 to 1080 nm and a zoom optics (OTZ-5 VR, Laserline GmbH), focusing the laser radiation vertically on a 16 cm (coating width) × 17 cm (length) spot of the coated foil. The laser voltage and intensity were adjusted through a control panel. A thermographic camera (Xi 400, Optris GmbH, Berlin, Germany) recorded the laser-drying process. The videos were analyzed using the software PIX Connect (Optris GmbH, Berlin, Germany). For convection drying, an oven with two heating zones and a total length of 2.1 m was used. The temperature of each heating zone was individually adjustable between 25 °C and 160 °C.

The process parameters of the coating and drying process are described in Table 2.

**Table 2.** Process parameter sets for coating as well as convection and laser drying.

| Process Parameters | Convection | | | | Laser | | | |
|---|---|---|---|---|---|---|---|---|
| | I | II | III | IV | V | VI | VII | VIII |
| Web speed (m/min) | 1.3 | 1.3 | 1.8 | 1.8 | 0.8 | 0.8 | 1.0 | 1.0 |
| Pump speed (rpm) | 220 | 220 | 290 | 290 | 120 | 120 | 140 | 140 |
| Wet film thickness (μm) | 160 | 160 | 160 | 160 | 160 | 160 | 160 | 160 |
| Temperature heating zone 1 (°C) | 150 | 160 | 150 | 160 | <25 | <25 | <25 | <25 |
| Temperature heating zone 2 (°C) | 130 | 140 | 130 | 140 | <25 | <25 | <25 | <25 |
| Laser voltage (V) | - | - | - | - | 0.7 | 0.8 | 0.85 | 0.95 |
| Laser intensity (W/cm$^2$) | - | - | - | - | 1.894 | 2.165 | 2.300 | 2.571 |

The process speed was varied between 1.3 m/min and 1.8 m/min for convection drying as well as 0.8 m/min and 1.0 m/min for drying via laser. The pump speed was individually adjusted according to the slurry rheology such that the wet film thickness was 160 μm for every process parameter set. When examining convection drying, the heating zones one and two of the oven were either heated to 160 °C and 140 °C, or 150 °C and 130 °C, respectively. While laser drying, the oven was not heated and exhibited a temperature below 25 °C. The laser voltage to control the laser was varied between 0.7 V and 0.8 V for a web speed of 0.8 m/min. When coating and laser drying with a process speed of 1.0 m/min, the laser voltage was differentiated between 0.85 V and 0.95 V. The resulting laser intensities for each process parameter set can be obtained from Table 2.

The variation in the process parameters aims at determining the influence of the drying technology, web speed and laser intensity on the quality of the anodes.

### 2.3. Quality Analysis Methods

The laser-drying process was characterized using thermography. Manufactured silicon–graphite anodes were analyzed in regard to the drying method, process parameters and material compositions as well as their effect on the electrode's quality.

Adhesion between the substrate and the coating was examined to evaluate the quality of the electrodes. Round samples of the anodes ($d_{sample}$ = 46 mm) were punched out and tested in a universal testing machine (5944 Single Column Table Top, 5940 Series, Instron GmbH, Darmstadt, Germany) with a testing software (Bluehill Universal, Instron GmbH, Darmstadt, Germany) and two orthogonal stamps as a testing tool. The stamps were covered with double-sided adhesive tape (Double-sided Universal, Tesa SE, Norderstedt, Germany) and the sample was placed on the lower stamp with the coating facing downwards. The upper stamp was pressed on the anode with a force of 930 N. Subsequently, the stamps were pulled apart at a speed of 100 mm/min. The maximum tensile stress, at which the delamination of the active material layer from the copper foil occurs, was measured.

Residual moisture of the anodes was investigated to compare the drying effects of the laser and the convection oven. Round samples of the anodes ($d_{sample}$ = 46 mm) were cut out and examined in a moisture analyzer (MA 50/1.X2.IC.A.WH, Radwag Waagen GmbH, Hilden, Germany). The device contains a scale and a halogen heating element. The sample's weight was determined. Subsequently, the anode was heated up to 150 °C such that residual moisture inside the sample is released and constantly weighed. The heating process was stopped when the weight of the sample did not change within a time frame of one minute (accuracy 1 mg). Using the weight difference of the sample before and after additional drying as well as the weight of the copper foil, the residual moisture was calculated.

Through-plane resistance ($R$) was measured to evaluate the electronic conductivity ($\sigma$) of electrodes using a universal testing machine (ZwickiLine Z 2.5 with 200 N Xforce load cell, Zwick/Roell GmbH & Co., KG, Ulm, Germany). Electrode sheets were laid between two copper stamps, applying different pressures. An ohmmeter (Resistomat Type 2316, Burster Präzisionsmesstechnik GmbH & Co., KG, Gernsbach, Germany) measures corresponding resistances $R$. The electronic conductivity is calculated via Equation (1), hereby $l$ represents the composite electrode thickness and $A$ the contact area of the stamps (6.45 cm$^2$). The procedure was adapted according to WESTPHAL et al. [8].

$$\sigma = \frac{l}{R \cdot A} \tag{1}$$

Thermogravimetric analysis (TGA) studies were conducted on an SDT Q600 (TA Instruments Inc., New Castle, DE, USA) at a heating rate of 10 K min$^{-1}$ in a temperature range of 30–600 °C with a nitrogen flow of 25 mL min$^{-1}$.

Scanning electron microscopy (SEM) was used to examine the surface of the electrodes and changes in particle morphology and composition. Round samples of the electrodes were cut out ($d_{sample}$ = 12 mm) and attached to the sample holder. The field emission gun (Skottky-type) was used for the measurement at an acceleration voltage of 3 kV. Different areas per electrode were analyzed in an Auriga CrossBeam workstation (Carl Zeiss AG, Oberkochen, Germany).

## 3. Results

### 3.1. Thermographic Analysis of the Laser-Drying Process

In Figure 2, thermographic images during the laser drying of silicon–graphite anodes containing 10 $w_{solid}$-% silicon at different parameter sets are shown. The characteristic drying profile is obtained for all material compositions (5, 10 and 20 $w_{solid}$-% silicon content).

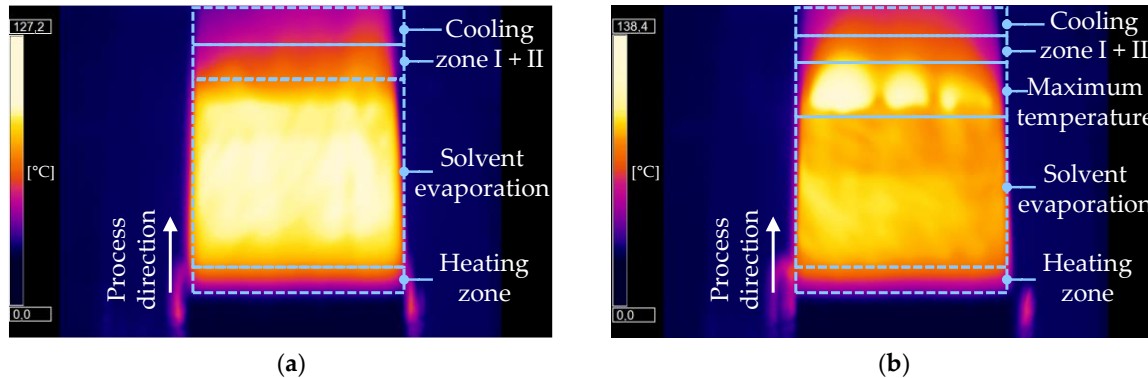

**Figure 2.** Thermographic monitoring of the laser-drying process for silicon–graphite anodes containing 10 $w_{solid}$-% silicon (**a**) at process parameter set V (low laser intensity), (**b**) at process parameter set VI (high laser intensity).

In the following section, the laser-drying process is described on the basis of Figure 2b, which shows the process parameter set VI. When laser radiation reaches the surface of the anode, the drying process starts. First, the material is heated from ambient temperature (<25 °C) to 80 °C in a short heating zone. As it ranges only a few centimeters, the anode experiences a rapid temperature rise. In the following zone, the electrode's temperature increases slightly to approximately 90–100 °C. Furthermore, the solvent starts to evaporate from the surface and is removed from the coating layer. This results in the consolidation of the electrode, with active material particles approaching each other and causing film shrinkage [9]. The solvent evaporation zone is characterized by its great length and the consistency of temperature across the area. This is due to the required evaporation heat, implying that the total energy input is needed for the evaporation process of the solvent. Consequently, the temperature on the surface of the anode as well as the drying rate are constant [10]. Once film shrinkage is finished and the final height of the coating is reached, capillary transport induces the emptying of solvent-filled pores within the anode's microstructure [11]. Pore emptying is characterized by a smaller solvent reduction, and thus, a decreasing drying rate [9]. During this drying step, the zone of maximum temperature is entered. As less energy is required for solvent removal, the anode's temperature rises to peak temperatures above 120 °C. It is observed that the temperature profile within this area varies greatly and spots with lower temperatures are observed as well over the course of the drying process. Finally, the dried electrode is not further exposed to laser radiation and cooling through the introduction of ambient air. The first cooling zone exhibits temperatures in the range between 65 °C and 80 °C, while the second shows temperatures between 50 °C and 65 °C.

Figure 2a shows the laser-drying process at process parameter V, operating at a lower laser intensity compared to Figure 2b. As previously described, the electrode experiences a heating zone where the material reaches temperatures up to 80 °C. The zone of solvent evaporation displays similar constant temperatures of 90–100 °C. In a difference to higher laser intensities, laser drying at process parameter set V does not exhibit a zone of maximum temperature and directly enters the cooling zones with a slightly lower temperature range. As a result, peak temperatures above 100 °C are not reached. Consequently, when decreasing the laser intensity, lower temperatures, or even a complete absence, are noted within the zone of maximum temperature.

### 3.2. Electrode Quality

In Figure 3, manufactured silicon–graphite anodes dried by convection (a) and by laser (b) are depicted.

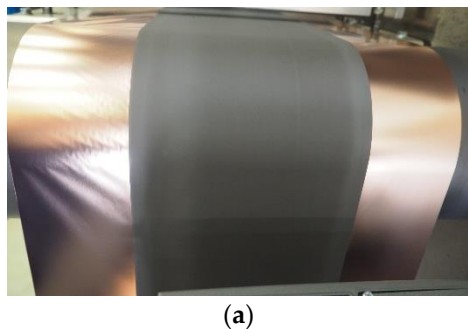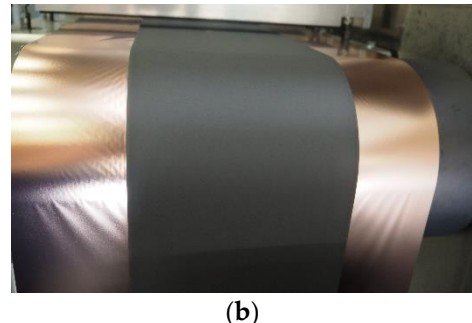

| (**a**) | (**b**) |

**Figure 3.** Silicon–graphite anodes (**a**) dried by convection, (**b**) dried by laser.

For both convective- and laser-dried electrodes, no cracks, conspicuous irregularities or damage on the surface can be detected. It is observed that all anodes manufactured at different process parameters show a similar optical appearance. As a result of a reduced graphite content, anodes are slightly brightened with increasing silicon content.

### 3.2.1. Adhesion

The adhesive forces for convection-dried anodes with varying silicon content are shown in Figure 4. No result was obtained for a 20% silicon anode at process parameter set III as the electrode was not sufficiently dried. After leaving the oven, the anode coating was still wet and could be scraped off the copper foil by a finger. All other anodes were sufficiently dry and further examined.

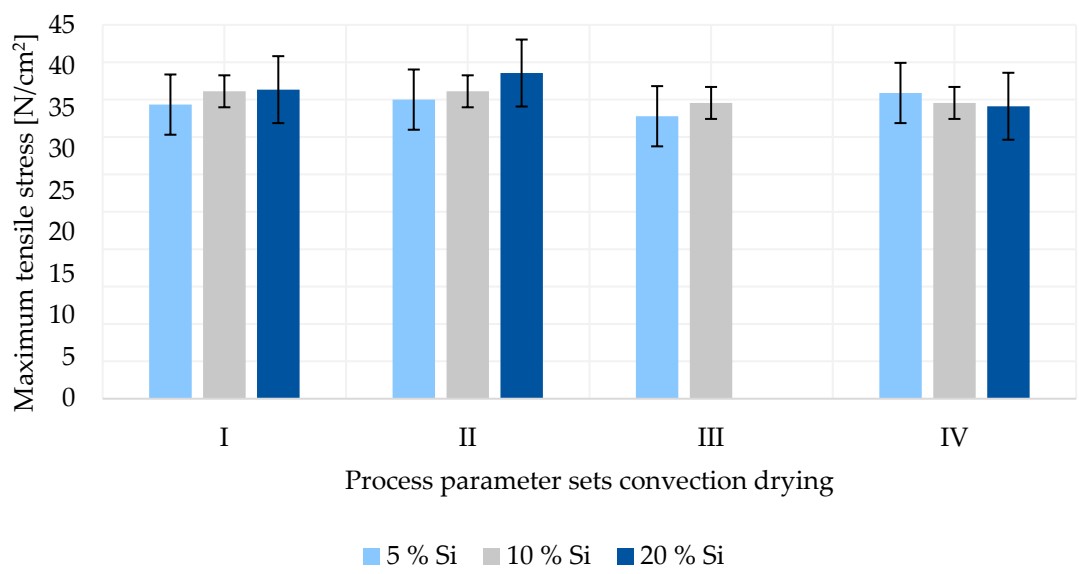

**Figure 4.** Adhesion force of silicon–graphite anodes dried by convection.

Figure 4 shows that varying the temperature of the heating zones of the oven between 150/130 °C (parameter I and III) and 160/140 °C (parameter II and IV) has no significant effect on the adhesive force of the produced anodes. Additionally, no difference is shown when increasing the web speed from 1.3 m/min (parameters I and II) to 1.8 m/min (parameters III and IV). Furthermore, it can be observed that with rising silicon content, the adhesion force slightly increases for parameter sets I to III. Only for parameter set IV is a lower maximum tensile stress measured with increasing silicon content. It is noted that the differences between the measurements are all within the standard deviation. Therefore, the described effects are not significant and not further considered.

In Figure 5, the adhesion forces for process parameter sets V to VIII for different silicon–graphite laser-dried compositions are displayed. It shows that with increasing

laser intensity at constant web speed (parameter VI > V and VIII > VII), the adhesion strength declines regardless of the material composition. The adhesion strongly depends on the binder materials and their homogenous distribution in the active material layer [12]. Low adhesion can be attributed to few binder particles at the silicon–graphite/copper foil interface [13]. The microstructure of the electrode, as well as the particle distribution, is formed during the drying process [14]. As the drying begins, active material, binder and carbon black particles are evenly distributed among the wet coating layer. The solvent evaporates, leading to the shrinkage of the film and particles forming a porous structure until the final thickness is reached. With solvent remaining between the particles, capillary transport mechanisms start to dominate the drying process. Consequently, not only the solvent, but also binder particles are transported to the surface. This phenomenon leads to a binder concentration gradient along the thickness of the electrode, thus resulting in the delamination of the coating and poor adhesion of the electrode [15,16]. It can be observed that a higher energy input into the material through a higher laser intensity leads to a stronger binder particle migration and therefore worsened adhesion forces. When comparing the web speeds of 0.8 m/min (parameter V and VI) and 1.0 m/min (parameter VII and VIII), no significant effect of the web speed on the maximum tensile stress can be found.

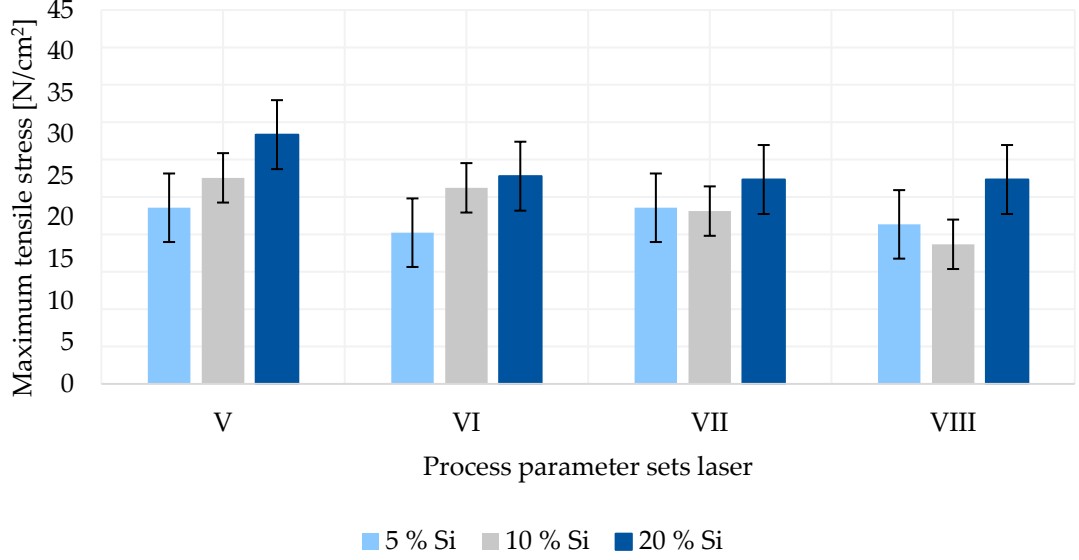

**Figure 5.** Adhesion force of silicon–graphite anodes dried by laser.

Furthermore, Figure 5 shows that an increase in the silicon content results in a higher adhesion force for laser-dried anodes. An exception is the 10 $w_{solid}$-% silicon anode at process parameter VIII. The abnormal measurement could originate from the electrode manufacturing or the adhesion testing procedure. Silicon may show a lower degree of absorption compared to graphite. Consequently, with a higher silicon content, less energy through electromagnetic radiation is absorbed by the coating during the laser-drying process. Lower energy input results in slower binder transport to the surface, leading to more uniform particle distribution and better adhesion between the substrate and the active material layer.

From Figures 4 and 5, it can be obtained that the adhesion for laser-dried anodes is slightly lower than for ones dried via convection. This phenomenon can be observed for anodes of all silicon contents and for all process parameter sets. Comparing the set-up for convection and laser drying, it can be noted that the laser drying time is reduced by over 80%, and thus, the drying rate is strongly increased. Higher drying rates may yield to lower adhesion, as the binder particles do not have sufficient time to diffuse back in the direction of the coating/foil interface for a homogeneous distribution [17].

### 3.2.2. Residual Moisture

The residual moisture values of anodes containing 10 $w_{solid}$-% silicon regarding the different process parameters for convection and laser drying are depicted in Figure 6. As all silicon–graphite anodes with varying silicon content show similar results, different material compositions are not displayed.

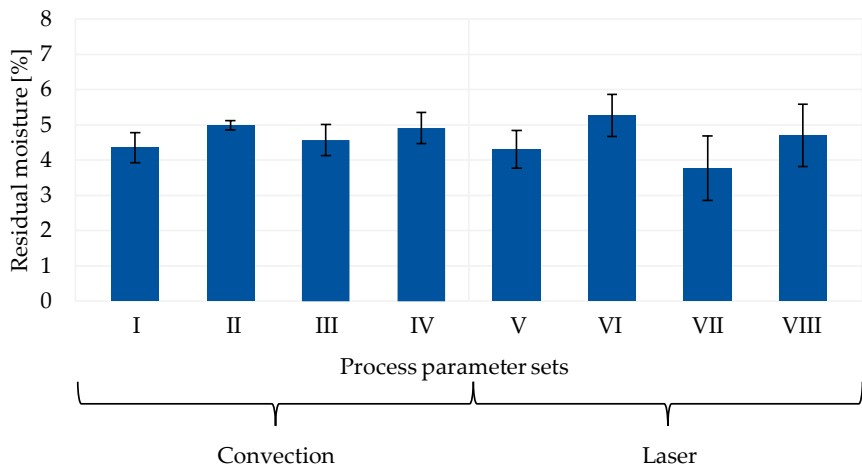

**Figure 6.** Residual moisture of convection- and laser-dried anodes with 10 $w_{solid}$-% silicon content.

It can be obtained that all samples exhibit a residual moisture between 3.7% and 5.2%. Anodes exposed to a lower energy input during drying do not show a significant larger moisture content in comparison to electrodes dried with a higher heating temperature or laser intensity (parameter I vs. II, III vs. IV, V vs. VI, VII vs. VIII). Furthermore, increasing the web speed, and thus reducing the drying time while applying the same heating temperature, does not lead to a change in the residual moisture content for convective-dried anodes (parameter I vs. III, II vs. IV). Comparing convection and laser drying of electrodes, no significant effect of the drying method on the remaining moisture is detected. Figure 6 shows a large standard deviation for most of the samples, implying that the measurements between similar specimens greatly differ. As the residual moisture testing method is not within the production line, a time frame of more than 30 min was required to prepare the samples and conduct the measurements. With normal ambient conditions in the experimental setting, anodes reabsorb moisture from the air, which diffuses into the electrode structure [18]. Therefore, it is suggested that residual moisture values adjust after drying through re-moisturization, and thereby approach each other, diminishing the effects of different process parameters and drying methods on the moisture content.

### 3.2.3. Electronic Conductivity

The electronic conductivity $\sigma$ within composite electrodes depends on the intrinsic conductivity of all electrode components as well as their distribution within the coating. The through-plane conductivity of composite electrodes dried with varying silicon content and varying drying conditions are shown in Figure 7.

Due to a lower intrinsic electronic conductivity of Si in comparison to graphite, higher silicon contents in silicon–graphite anodes result in a lower electronic conductivity of composite electrodes as shown in Figure 7. Comparing selected different drying procedures, convection-dried electrodes (parameter II) show higher electronic conductivities in comparison to laser-dried electrodes (parameter V). Furthermore, the laser intensity affects the electronic conductivity as higher laser intensities (parameter VI) result in lower $\sigma$. This effect is also attributed to binder migration. High ratios of insulating binder result in increased resistances at the composite electrode surface and thus to a lower overall electronic conductivity.

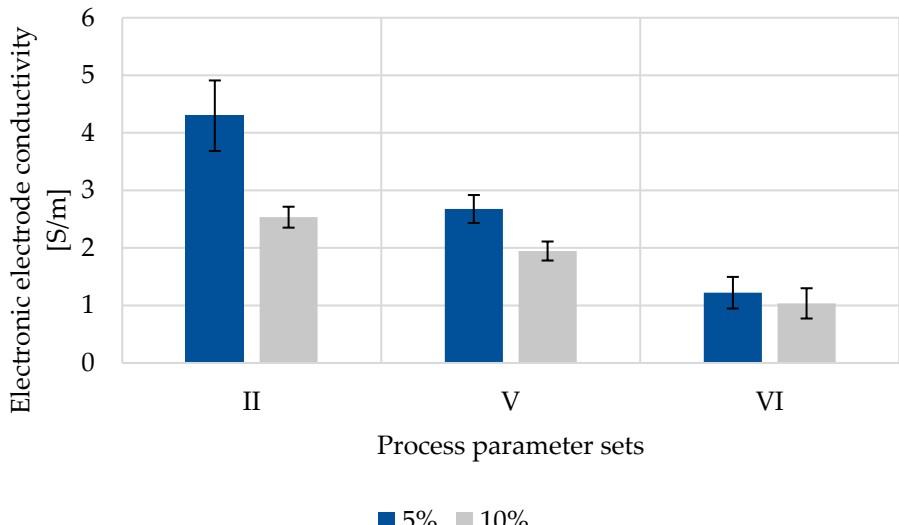

**Figure 7.** Through-plane electronic conductivity of composite electrodes with varying Si contents and under varying drying conditions at a contact pressure of 30.2 N cm$^{-2}$.

### 3.2.4. Thermal Stability

As shown above, different drying procedures affect the binder homogeneity within the composite electrode. Therefore, thermogravimetric analysis (TGA) is used to investigate the thermal stability of the binder after different drying procedures. Within the selected temperature range, CMC decomposes at 250 °C and SBR at 400 °C, yielding mass loss as shown in Figure 8. No significant differences are obtained for the different drying procedures (Figure 8a). Therefore, it is concluded that even the most harshly herein applied laser intensities (parameter VIII) do not deteriorate the binding abilities of CMC and SBR within composite electrodes.

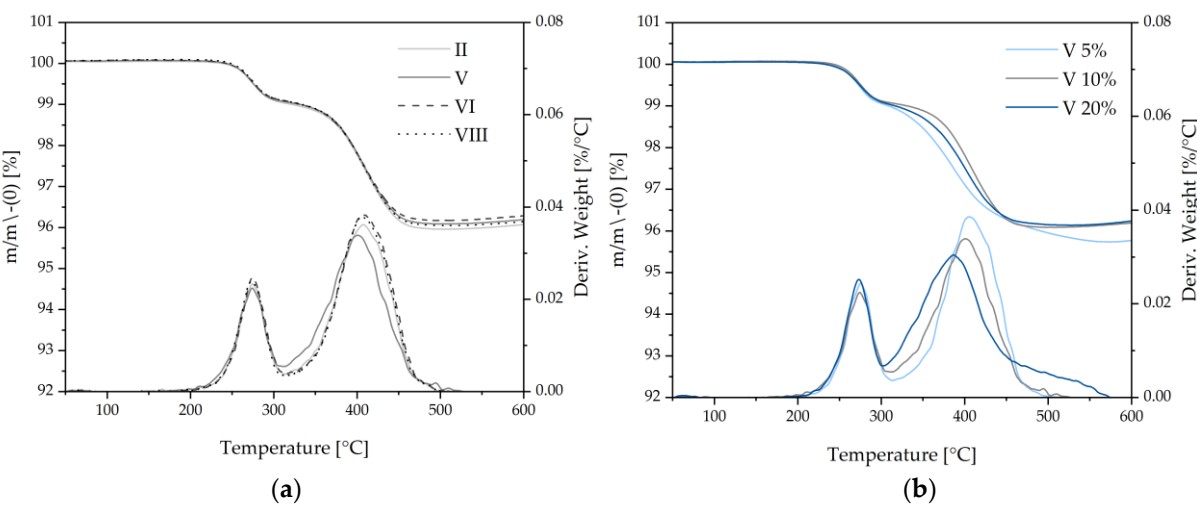

**Figure 8.** TGA investigations of silicon–graphite anodes (**a**) dried under four different procedures with 10 $w_{solid}$-% Si content and (**b**) dried under similar laser procedures with 5, 10 and 20 $w_{solid}$-% Si content.

Higher silicon contents in silicon–graphite anodes result in a shift in the decomposition temperature of SBR towards lower temperatures as shown in Figure 8b. This may be due to the worse adsorption of SBR on the silicon surface in comparison to graphite, resulting in decreased binding abilities of SBR within high silicon content composite anodes [19].

### 3.2.5. Morphology

Figure 9 shows SEM images of the electrodes with 10 $w_{solid}$-% silicon manufactured with convection or laser drying at a magnification of 10,000 (upper images) and 250 (lower images).

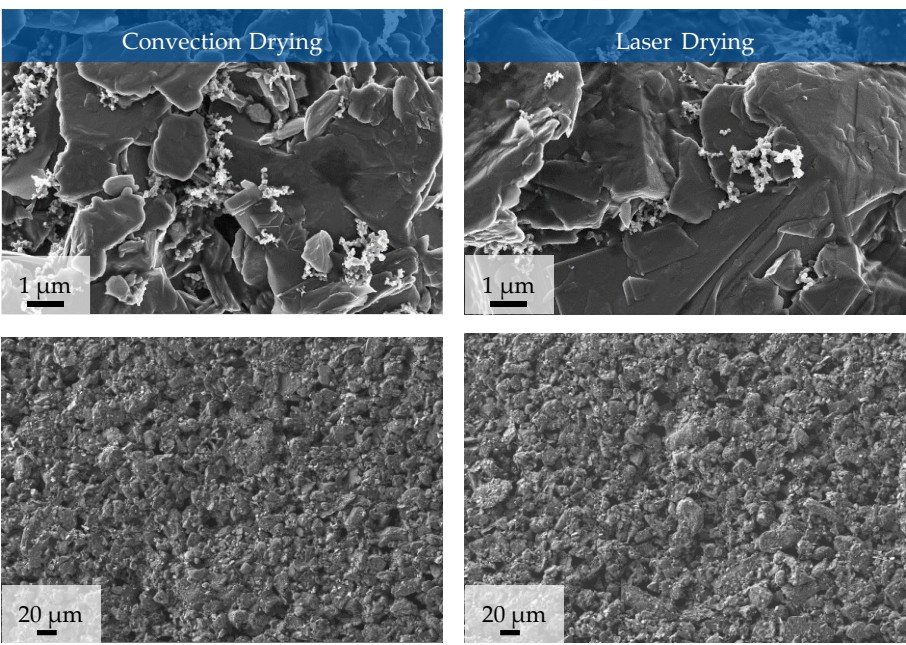

**Figure 9.** SEM images of anodes containing 10 $w_{solid}$-% silicon at process parameter set II (convection drying) and at process parameter set V (laser drying).

A detailed analysis of different spots on the surface of the electrodes shows that there are no differences visible in the morphology in the composition of the electrode surfaces after convection drying and laser drying. The figure shows the surface of the silicon–graphite anodes. The distribution of conductive additive on the active material is largely homogeneous, with small agglomerates visible. Smaller particle sizes and thus more edges ensure that more conductive additive adheres in these areas. The images of the electrodes in smaller magnification (Figure 9 lower images) show that no cracks develop due to drying.

### 4. Discussion

It was shown that silicon–graphite composite electrodes can be manufactured using laser-drying technology. The drying process was analyzed by thermography and classified into characteristic drying areas. The knowledge of the temperature profiles during laser drying can deepen the understanding of the occurring processes on a particle scale. It can be used to optimize the conceptional layout and design, such as the laser intensity and spot area.

The adhesion forces of laser- and convection-dried samples with varying silicon content at different process parameters were examined. Findings in the literature suggest an influence between process settings and the adhesion [15,20,21]. No significant effects were found for convection-dried anodes, which may be due to the marginal changes in the process parameters. It was observed that increasing the silicon content and lowering the laser intensity leads to higher adhesion for laser-dried anodes. These results are in accordance with other publications, suggesting that a higher energy input during drying promotes the rate of capillary transport of the solvent to the surface, and thus, binder migration [11]. Furthermore, the findings suggest that silicon–graphite anodes are highly suited to be dried via laser. Silicon contents above 20% or pure silicon anodes may be subjected for further investigations in regard to laser drying.

Residual moisture was evaluated for all samples and no difference was found between convection and laser drying, which has also been reported by VEDDER et al. [22]. It is suggested that the testing method to record residual moisture needs to be improved further. An approach can be the development of an inline measurement, capturing moisture values directly after drying within the production line. Additionally, manufacturing and testing activities under dry room conditions would prevent the re-absorption of moisture via ambient air. This effect needs to be considered when assessing the residual moisture of electrodes.

Through-plane conductivity measurement results also suggested that binder migration occurs during laser drying of the electrodes [11]. Furthermore, the effect is amplified for higher laser intensities.

Via TGA investigations it was shown that even the harshest herein applied laser drying conditions do not corrupt the binding abilities of CMC and SBR within the binder network. This further implies Si/C anodes are highly suitable for composite electrode laser drying.

Based on the SEM images, no differences in the composition of the electrode surface between convection drying and laser drying could be detected. Effects such as the elutriation of the binder or the increased agglomeration of the conductive additive could not be detected. These results indicate that the laser drying used here could replace convection drying without affecting the electrode surface quality.

Another approach that should be examined further is the use of a hybrid drying model, utilizing laser and convection in one concept. A laser system could pre-dry the electrode until film shrinkage is completed. Subsequently, a convection oven could be used to evaporate the solvent left in the pores of the anode's structure. As binder migration primarily takes place during pore emptying, a homogeneous distribution of binder particles may be accomplished in this setting through low heating temperatures in the heating zones of the oven [23]. Accordingly, this concept has the potential to combine the advantages of laser and convection drying, which should be proved in further investigations.

## 5. Conclusions

The investigation of silicon–graphite anodes with laser-based drying has demonstrated that a composite electrode can be processed with stand-alone laser drying. During the process, it was found that the measured temperatures during laser drying react very sensitively to the set laser intensity. Therefore, inline monitoring of the laser-drying process is useful to ensure that sufficient drying is achieved while avoiding high temperatures that would lead to binder decomposition. Comparing the settings for convection and laser drying, it can be seen that the drying time can be reduced by over 80% when using the laser technology, greatly increasing the drying rate. Due to the shorter drying time and more energy-efficient heat input, the process innovation of laser drying has the potential to provide significant cost savings in electrode manufacturing. However, other concepts, such as hybrid laser drying, should be investigated to further improve the electrode quality, e.g., the adhesion and electronic conductivity as a result of binder migration.

**Author Contributions:** Conceptualization, S.W. and L.G.; methodology, L.G., S.W., V.G. and R.T.; formal analysis, S.W., L.G., V.G. and R.T.; investigation, L.G., S.W., R.T. and V.G.; writing—original draft preparation, S.W., L.G., V.G. and R.T.; writing—review and editing, S.W., L.G., V.G. and R.T.; visualization, S.W., L.G., V.G. and R.T.; supervision, D.N., H.H.H. and M.B.; project administration, S.W. and M.B.; funding acquisition, D.N., H.H.H. and M.B. All authors have read and agreed to the published version of the manuscript.

**Funding:** This work is part of the Project IDEEL. This research was funded by the Federal Ministry of Education and Research, grant number 03XP0414D.

**Institutional Review Board Statement:** Not applicable.

**Informed Consent Statement:** Not applicable.

**Data Availability Statement:** Not applicable.

**Acknowledgments:** The integration of the high-power diode laser into the foil coating system at the eLab of RWTH Aachen University was kindly carried out with the technical support of the company Laserline GmbH.

**Conflicts of Interest:** The authors declare no conflict of interest. The funders had no role in the design of the study; in the collection, analyses, or interpretation of data; in the writing of the manuscript; or in the decision to publish the results.

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
