# Peer review of "Process and Material Analysis of Laser- and Convection-Dried Silicon–Graphite Anodes for Lithium-Ion Batteries"

_wevj, doi:10.3390/wevj14040087_

Round 1
Reviewer 1 Report
The work reported a new laser drying method to prepare silicon/graphite anodes. Some issues should be considered in their revision.
1. Do you measure and compare the electrochemical properties of the anodes based on different drying methods?
2. A low-magnification SEM image is suggested to provide for showing the overall surface quality of the electrode, such as any there any possible cracks.
3. Some work of LIBs, such as Rare Metals. 2021, 40(4):837–847,Part. Part. Syst. Charact. 2021, 38: 2100107, could enrich the background.
Reviewer 2 Report
The article „ Process and Material Analysis of Laser and Convection Dried Silicon-Graphite Anodes for Lithium-Ion Batteries “ is devoted to an important Li-ion battery production step - drying. Two methods have been compared, and conclusion has been made that the laser drying can potentially decrease the cost of the battery cell production. From my point of view the manuscript is interesting, well-structured and presents extensive experimental results. I would recommend to publish the manuscript after small minor changes.
English grammar, particularly usage of a and the articles, should be checked.
The abstract lists what has been done in the work, I would advise to present briefly main results obtained.
Results section lines 155-157 logically belong to the section 2.3. Quality analysis methods.
In the section 3.1 Thermographic analysis of the laser drying process is presented for 10%-silicon sample at the process parameters V and VI. It is not mentioned in the text, does the identical peculiarities of laser drying process are observed for 5% and 20% Si samples.
Page 10 Lines 330-332 “A detailed analysis of different spots on the surface of the electrodes shows that there is no difference in the composition of the electrode surfaces after convection drying and laser drying”. Figure 9 shows morphology of the samples, to make conclusion about differences in the electrode’s composition, EDX-map measurements should be given.
